# Abortion information-seeking experiences among women who obtained abortions in Kinshasa, DRC: Results from a qualitative study

Denise Ngondo[1], Celia Karp[2]*, Dynah Kayembe[1], Kisulu Samyonga Basile[1], Caroline Moreau[2,3], Pierre Akilimali[4], Suzanne O. Bell[2]

1 Department of Biostatistical Epidemiology, Patrick Kayembe Research Center, Kinshasa School of Public Health, Kinshasa, Democratic Republic of the Congo, 2 Department of Population, Family and Reproductive Health, Johns Hopkins Bloomberg School of Public Health, Baltimore, Maryland, United States of America, 3 Soins et Santé Primaire, CESP Centre for Research in Epidemiology and Population Health U1018, Inserm, Villejuif, France, 4 Department of Nutrition, Patrick Kayembe Research Center, Kinshasa School of Public Health, Kinshasa, Democratic Republic of the Congo

* celia.karp@jhu.edu

**Data Availability Statement:** The blinded data that support the findings of this study are publicly available via the Performance Monitoring for Action

## Abstract

Little is known about the process of seeking information related to abortion care options among women in the Democratic Republic of Congo (DRC). Understanding how women obtain information can help identify opportunities for intervention to increase awareness and use of safe pregnancy termination options. Using qualitative data collected from women in Kinshasa, DRC who reported having an abortion in the last 10 years, this study aims to determine how women navigate obtaining information about their options for abortion and the role of their social network in their information-seeking processes. Data for this analysis come from a mixed-method study of abortion in Kinshasa conducted from December 2021 to April 2022. Fifty-two qualitative interviews followed a structured interview guide, including open-ended questions and probes, developed by a multidisciplinary team of researchers in Kinshasa and the United States. Inductive thematic analysis was conducted using Atlas.ti, and a thematic analysis matrix was used to describe the major themes and subthemes. Thematic analysis revealed two main themes with nested subthemes. The first and most salient theme highlighted the highly selective and narrow information search process women engaged in, involving no others or very few individuals (e.g., partners, women in one's community, or providers) that the pregnant woman chose strategically. The second theme revealed the heterogeneous and often stigmatizing nature of these interactions, including attempts at deterrence from many sources and information of varying completeness and accuracy. While the recent liberalization of the abortion law in the DRC is essential to improve access to safe abortion, public health gains will not materialize unless they are accompanied by community-level actions to raise awareness about the legality and availability of safe abortions services, including medication abortion pills for safe self-managed abortion.

(PMA) project, upon reasonable request. Requests for data access can be made via https://datalab.pmadata.org/ with information about the researcher's planned use of the data. Public deposition of these data would breach compliance with the protocol approved by the research ethics board, as this was not explicitly indicated in the informed consent process and would violate the study's approved ethical procedures for participant engagement and data management.

**Funding:** Funding for this study was provided by the David and Lucile Packard Foundation (grant number 2021-72199 - SOB) and the Bill & Melinda Gates Foundation (grant number IVN009639 - PA). Funders were not involved in any aspect of the study design, data collection and analysis, nor interpretation and writing of the manuscript.

**Competing interests:** The authors have declared that no competing interests exist.

## Introduction

Induced abortion was prohibited in the Democratic Republic of Congo (DRC) until 2018, when the country officially decriminalized it by codifying their ratification of the African Union's Maputo Protocol into law [1, 2]. Abortion can now legally be accessed through 14 weeks' gestation in cases of sexual assault, rape, incest, fetal abnormalities, and when continuing the pregnancy endangers the mental or physical health or life of the pregnant woman. While permissive abortion laws are generally associated with greater abortion safety [3], many abortions occurring in the DRC remain unsafe due to inadequate healthcare services [4] and stigmatization, leading many women to seek care outside the formal healthcare system [5].

There were an estimated 105 (95% CI 79–132) abortions per 1,000 women aged 15–49 in Kinshasa, the capital of DRC, in 2021, most often involving procedural abortion and private facilities [6]. In addition, 20% of abortions in Kinshasa involved self-managed medication abortion [6], a safe alternative to facility-based care [7]. However, approximately one-third of abortions remain unsafe, involving non-recommended methods (i.e., anything other than procedural abortion or medication abortion) and non-recommended sources (i.e., non-clinical sources for procedural abortion) [6], increasing the risks of maternal morbidity and mortality. Prior research in Kinshasa before liberalization of the abortion law suggested nearly half of women who had an unsafe abortion experienced moderately severe complications [8]. Complications of unsafe abortion are a leading cause of maternal mortality in sub-Saharan Africa, responsible for an estimated 10% of maternal deaths in the region [9]. In Kinshasa, one small study found that 15% of patients receiving care at an obstetrics and gynecology emergency room were treated for complications of unsafe abortion, 6% of whom died [10].

The continued reliance on unsafe abortion, despite the recent liberalization of the abortion law in DRC, coupled with the increasingly widespread availability of abortion pills, suggests other factors continue to preclude access to safe abortion in this context. As of 2021, fewer than one in five (19%) women in Kinshasa knew that the law allows abortion in certain circumstances [11]; this proportion was even lower among adolescents, women with a low level of education, and those from the poorest households [11]. This lack of awareness can limit perceived options for information care seeking if women fear legal sanctions for pursing an abortion.

Understanding how women obtain information about where to access abortion in a setting characterized by limited abortion-related knowledge and access to care is essential in reducing morbidity and mortality due to unsafe abortion. Lack of knowledge about abortion methods and sources, which often depends on an individual's residence, age, and education, presents an initial obstacle to accessing safe abortion care [12]. This extends to knowledge of medication abortion specifically, which has been found to be higher among younger women [13]. Incomplete and inaccurate information often leads to misperceptions about risks, efficacy, and safety, deterring women from seeking care within formal healthcare settings [14]. Despite inadequate knowledge, research has found that women often limit their discussion of their abortion decisions to few, select people or keep the decision to abort entirely a secret, out of fear of stigmatization and potential rejection from family, friends, and partners [15–17]. Fear of stigmatization may be greater in a highly Catholic country like the DRC given the church's prominent role in society and its opposition to abortion. Women who decide to engage others in their abortion care trajectories generally rely on few informants from their communities who often provide incomplete information [18], exposing them to potential negative health consequences [12, 19, 20].

Little is known about the process of seeking information related to abortion care options among women in the DRC. Using qualitative data collected from women in Kinshasa who reported having an abortion within the last 10 years, this study aims to determine how women navigate obtaining information about their options for abortion and the role of their social network in their information-seeking experiences. Understanding how women obtain information when seeking an abortion in Kinshasa can help identify points of interventions to increase awareness and use of safe pregnancy termination options.

## Methods

### Overview of research methods

Data for this analysis come from a mixed-methods study of abortion in Kinshasa and Kongo Central, DRC conducted in 2021–2022. The study was conducted as part of the Performance Monitoring for Action (PMA) project, which implements population-based surveys to monitor key family planning and reproductive health indicators in nine countries throughout sub-Saharan Africa and south Asia. Additional information about the PMA project, including the methodology, questionnaires, and data, is available at www.pmadata.org/data/survey-methodology. In the DRC, PMA is led by the Kinshasa School of Public Health with technical support provided by the Johns Hopkins Bloomberg School of Public Health. Ethical approval for this study was obtained from the Institutional Review Board at the Johns Hopkins Bloomberg School of Public Health (#14590) and the Comité d'Éthique at the Kinshasa School of Public Health (#ESP/CE/159B/2021).

The initial phase of the PMA DRC abortion study included a cross-sectional survey using a multi-stage cluster random sampling design to identify a representative sample of women aged 15–49 years living in Kinshasa. A total of 2,326 participants provided informed consent and completed a face-to-face survey that collected information on their sociodemographic profile, reproductive history, contraceptive practices, and abortion history. Women in Kinshasa who completed the quantitative survey [6] and reported an abortion in the last 10 years were eligible to participate in the qualitative follow-up interview. This study uses the data from these qualitative interviews.

Altogether, 93 women reported having an abortion between 2015 and 2021 in the quantitative survey and agreed to be contacted for a follow-up interview about their abortion experience. Women were contacted by telephone according to a script approved by the ethics committee. From the 93 eligible participants, interviewers first attempted to contact those who had the most recent abortions, continuing to contact women until reaching the goal of 50 completed interviews. In total, interviewers attempted to contact 65 women, among whom 52 women provided written informed consent to participate in an in-depth interview about their abortion experience and allow a digital recording of the interview. The interviews were conducted at a day, time, and place of the woman's choosing.

Nine qualitative interviewers–all of whom resided in Kinshasa and had prior data collection experience via the PMA project–conducted the interviews after completing a four-day training about research ethics, informed consent, qualitative research approaches, and interviewing techniques. The interviews, conducted in Lingala, lasted 30–75 minutes. Interviews followed a structured interview guide, including open-ended questions and probes, developed by a multi-disciplinary team of researchers in Kinshasa and the United States. The guide was pre-tested during the interviewer training and addressed the following topics: 1) local abortion terminology used by women, 2) information-seeking practices for abortion options, 3) decision-making processes, and 4) abortion care trajectories. Interviewers also asked women to reflect on the abortion laws in the country and share any recommendations for improving the quality of

care they received during their abortion experiences. The interviewers took notes to complement the audio recordings of each interview. All transcripts were translated verbatim into French by two transcribers, with random checks for quality of translation. Confidentiality and anonymity were guaranteed during each participant's informed consent process, and participants were not identified by name or any other identifier in study materials. Audio recordings were immediately downloaded onto a password-protected computer, then erased from the recording device. The interviewers' notes were archived in secure cupboards in locked offices on the premises of the local DRC team. No personal information about respondents was included in the interviewers' notes. All women were reimbursed $2 US for time spent in the interview and to cover transportation costs. Data were protected in locked, password-protected computer files and were de-identified for analysis. Only the project team had access to the raw data. The recordings will be archived for five years, after which they will be destroyed.

## Analysis

The study team used Atlas.ti 22 for data management, coding, and analysis. First, members of the study team reviewed all interview transcripts in French and generated memos for each transcript, recording any salient ideas, questions, or important notes about the transcript and/or participant. Next, a team of six researchers (three based in the DRC and three based in the US) developed a common codebook using open coding techniques—i.e., identifying potential codes from three transcripts, followed by discussions among the research team members to identify and generate a draft codebook. The coding approach followed a paradigm developed by Strauss and Corbin (1990) to identify terminology and definitions for coding subcategories (e.g., phenomenon, causal conditions, strategies, consequences, context, and intervention conditions related to abortion information seeking) [21]. Next, we conducted axial coding—i.e., the development of a coding structure and a final codebook that was used for the remaining transcripts. Once the final codebook was established, each researcher coded approximately 10 transcripts, and then each coded transcript was reviewed by another team member to ensure consistency among coders and completeness of codes. During the coding phase, memos were recorded to explore any emerging research questions or ideas. Coders also noted any inconsistencies in participants' accounts of their abortion experience so that discordant dimensions of care could be explored.

After coding was finalized, we conducted an inductive thematic analysis and developed a coding report in Atlas.ti to export a complete list of coded quotes with the relevant codes for our analysis. We then developed a matrix to identify and describe the major themes and subthemes. The matrix included relevant quotes illustrating each theme or subtheme, including participant characteristics (e.g., age, marital status, parity) for each quote. The researchers also reviewed the transcripts directly, focusing on sections of the interviews that asked respondents how they sought information about their abortion options and whom they used in the process, to identify potential quotes, subthemes, and themes that had not been captured in the original coding report. Inductive thematic saturation techniques were used to achieve saturation of themes and sub-themes. Research team members focused on identifying newly emergent codes and themes, instead of the completeness of the existing codes and themes, at each research team meeting until no new codes or themes emerged [22]. Rigor was maintained throughout the research process through weekly research team meetings to discuss challenges, or questions regarding the interpretation of quotes, and ideas related to conceptual linkages between key themes, which helped generate the final thematic matrix.

## Results

### Participant characteristics

More than half of the women interviewed were younger than 30 years old and had never been married (Table 1). Nearly 7 out of 10 women had completed secondary school. Thirty-six (69.2%) women had at least one child. Over half of women received a procedural abortion (51.9%), the next most common method being pills other than medication abortion pills

**Table 1. Characteristics of qualitative sample of reproductive-aged women in Kinshasa, DRC who had a recent abortion.**

|  | n | % |
|---|---|---|
| **Age** |  |  |
| 18–19 | 5 | 9.6 |
| 20–24 | 10 | 19.2 |
| 25–29 | 20 | 38.5 |
| 30–39 | 15 | 28.9 |
| 40–49 | 2 | 3.9 |
| **Residence** |  |  |
| Slum | 39 | 75.0 |
| Non-Slum | 13 | 25.0 |
| **Marital status** |  |  |
| Never married | 28 | 53.,9 |
| Married | 10 | 19.2 |
| Living with a partner | 9 | 17.3 |
| Divorced / separated | 5 | 9.6 |
| **Education** |  |  |
| Primary | 1 | 1.9 |
| Secondary | 36 | 69.2 |
| Tertiary | 25 | 28.9 |
| **Parity** |  |  |
| 0 | 16 | 30.8 |
| 1–2 | 26 | 50.0 |
| 3 and more | 10 | 19.2 |
| **Abortion year*** |  |  |
| 2018 or earlier | 14 | 26.9 |
| After 2018 | 38 | 73.1 |
| **Abortion method**** |  |  |
| Procedural abortion | 27 | 51.9 |
| Medication abortion pills | 11 | 21.2 |
| Other pills | 18 | 34.6 |
| Injection | 10 | 19.2 |
| Traditional or "other" methods | 7 | 13.5 |
| **Abortion source**** |  |  |
| Public facility | 16 | 30.8 |
| Private facility | 23 | 44.2 |
| Pharmacy | 22 | 42.3 |
| Other source | 5 | 9.6 |

*: One woman missing information ** Women could report more than one abortion method and source thus these numbers/percentages may not sum to total 52/100%.

(34.6%). Most women visited private facilities (44.2%) or pharmacies (42.3%) to obtain their abortion.

Thematic analysis revealed the intersection of two main themes related to care-seeking behaviors and quality of interactions. The first and largest theme on care-seeking behaviors highlighted the highly selective and narrow information search process women engaged in, involving no others or very few individuals (e.g., partners, close female networks, or health care providers) that the pregnant woman chose strategically. The second theme on the quality of these interactions revealed the heterogeneous and often stigmatizing nature of these conversations, including attempts at deterrence and information of varying completeness and accuracy.

## Care-seeking as a selective process

Women's search for abortion care information mostly depended on their sense of social safety, as abortion remained widely stigmatized and potentially punishable by law. Confidentiality guided their paths, favoring self-reliance if women had prior knowledge, extending to partners if they were supportive or activating trusted female networks (family, friends, or neighbors) if women had no option but to seek external support.

## An autonomous course

Respondents who already knew about methods and places of abortion care—either through their education or medical training, their circle of acquaintances, or because they had had a prior abortion—said they did not feel compelled to solicit information from their social network, often managing the situation on their own to avoid the risk of disclosure.

As one 33-year-old woman shared about her ability to navigate her abortion options due to her profession, "*Yes, I already knew some methods . . . It's because I, myself, am a medical professional [health care provider] . . . As soon as I arrived at the pharmacy, and I had bought the medication . . .I had written my prescription myself and the pharmacist had served me.*" (Divorced woman, no children).

Other women shared how their prior experiences with abortion shaped their knowledge and pursuit of specific abortion methods. A 28-year-old woman stated, "*Yes (I knew the methods of abortion) since I had already done it once. I know that they give a medicine to put vaginally and another to inject in the buttock. Another method, they use instruments that open the cervix and then circle the blood.*" (Separated woman, one child).

Women in these scenarios were not obligated to disclose their abortion to anyone in order to learn about methods and sources for seeking care. This limited potential social sanctions, sometimes at the expense of more accurate and complete abortion information from other sources.

## Activation of the social network

Many respondents, however, possessed no prior knowledge about abortion, requiring them to "activate" their personal networks. Women were very selective about who they approached, often limiting their interactions to a trusted individual, whether a partner, sister, or friend, to maintain secrecy throughout their abortion care trajectory. The nature of these interactions resulted in some women being dependent on the knowledge of only one confidant to protect their reputation.

**Role of the partner.**   Partners' involvement in women's information-seeking about abortion was often crucial, acting as an intermediary to seek out accessible abortion methods and sources within their network. While some men shared this information with their partners, so

that the woman could decide on a course of action, others made arrangements for the abortion directly without involving the woman. In such cases, women described following their partners' instructions for accessing their abortion without prior knowledge of care options and modalities.

One respondent described her partner's role in sharing information and options about abortion, *"It was my boyfriend who knew about abortion methods such as curettage, taking the drugs. He had explained his methods to me and asked me to choose between the curettage and taking the medication."* (18-year-old woman, never married, no children).

Far more common, however, was the partner's facilitation of connections to providers, booking of abortion appointments, and transport for the woman to receive care. As one 27-year-old respondent described, "*My husband had accompanied me to a health center where a brother worked with whom we prayed, he was a doctor there and my husband had called his brother who is also a doctor but who worked in another health center to come and get me an abortion. It was my brother-in-law who had given me the medicine [misoprostol] to open the cervix."* (Divorced/separated woman, 2 children).

Similarly, another young woman described the key role her partner played in helping her access care from a physician: *"They (the doctor, my boyfriend, and his friend) had a three-way conversation, and when they were done, he (the doctor) took me to his center, which was right next to the bar."* (24-year-old woman, never married, no children).

In some cases when women engaged their partners in their information-seeking processes, they became excluded from the process, as their partners exerted control and authority over their abortion care trajectories. For some respondents, this leadership in the process was welcome, as one young respondent shared, *"In fact he (my spouse) had taken me there, I didn't know . . .he was the one who knew that doctor . . .when we arrived, he (my spouse) had first spoken face to face with the doctor, I hadn't taken part in that conversation . . .I didn't have any methods in mind with me, I didn't know any methods."* (24-year-old woman, never married, no children).

**Role of women's female networks.** Information-seeking about abortion also often involved engaging with women's close female networks, emphasizing relationships of trust to minimize the risk of disclosing the abortion. For example, women seeking information about their options for abortion care sought out their sisters, close female friends, or female neighbors to learn about methods, providers, and, more generally, how to navigate the healthcare system discreetly—without being recognized.

Women described their sisters as their preferred confidantes, given their shared interest in keeping the abortion secret for the sake of their family's reputation and their ability to activate a close social network of women, including individuals who may have had an abortion. Sisters often served as connectors, forging interactions between the woman seeking an abortion and others whom they trusted. As one respondent described, "*I had a little sister that we usually interacted with, and it was to her that I had told my story a little bit. She had told me that she had a classmate to whom she had introduced me who had asked me questions."* (31-year-old woman, never married, two children).

Women emphasized the importance of ensuring anonymity and safety in one's information-seeking process as a key motivation for confiding in sisters. A 36-year-old woman shared this experience saying, *"Everyone has someone they confide in, so for me it was my sisters that was the reason. Some people are not able to keep the secret and decide to spread the rumor everywhere. I knew that with them [my sister], I was safe."* (Married woman, four children).

Female friends were also often brought in as support because they either knew someone who had had an abortion or recently had an abortion themselves. Such women were described as trustworthy and reliable sources of information about which method to use, which provider

to contact, and where to seek care. As one woman discussed her own experience confiding in a friend, saying, *"I told her, 'Okay, but I don't know how it goes'. However, I have a classmate who has already done it and may know where to do it. She can guide me. Then she told me, 'No problem, I know an uncle who does it; he has already done it twice with me. . .He'll give you a shot and some pills to take'."* (28-year-old woman, never married, one child). Another respondent emphasized how the nature of her long relationship with her friend instilled a level of confidence in her friend's advice, stating: *"I have a friend who advised me to have an abortion. She is a childhood friend. She is already used to doing it in clinics . . .I knew that she would support me and that she could keep it a secret."* (27-year-old woman, never married, no children).

Female neighbors were also solicited, participating in the informal circulation of information in the community. *"I had gone to see a sister like that in the neighborhood, I had explained to her that I was already pregnant for about two months, and I don't want to use medicines at that age. I would like to go to the hospital for the curettage. She referred me to a hospital near Kyanza. I didn't know about curettage before . . .it was just the sister who talked to me and referred me to this method."* (27-year-old woman, never married, no children).

However, the involvement of female networks resulted in a risk of social retribution through disclosure—a situation reported by several respondents who were victims of their confidants' indiscretions. This led some women to conceal their condition (i.e., seeking abortion) by requesting information about the abortion on behalf of a third party to limit personal risk.

As a 19-year-old woman shared how her experience of information-seeking resulted in intimate partner violence, *"I didn't know anything. I had a friend who knew where they sell abortion drugs, and she took me there. That friend finally sent the information to my boyfriend's friends, and one day, my boyfriend caught me along the way. He beat me up. He asked me to give him his child that I had aborted. . .I think my friend there had no secrets, and she was the one who sold me to my boyfriend."* (Never married woman, no children).

Consequences of female network disclosure also extended to women's families. In some cases, women's trusted female friends disclosed their status as abortion-seekers to their families; as one respondent described, *"No, I didn't tell them (the family), it was my friend who found out I was pregnant, and she was the one who first told me that I wanted to abort it. So, she also went and told my family, 'Your child is looking to have an abortion'."* (23-year-old woman, married, one child).

Finally, women who lacked trust in their family, female friends, or partners shared how they would fabricate their own experiences as narratives of others to gauge others' perspective. For example, one woman shared: *"What do I usually do? I may have a problem that is beyond me, and I go to an old or mature person. . .tell him that I have a friend who is going through such and such a situation, and I criticize my fictitious friend and ask, 'How she should behave in the face of what has happened to her?' And, I already start to have a position in front of these discussions. At that moment, he won't know that I am the one who is going through this situation. That's what I had done."* (22-year-old woman, never married, no children).

**Role of health care providers.** To minimize the risk of disclosure and inaccurate information, some women went directly to health care providers, including pharmacists or health facility providers, to seek information and receive care. Health professionals were viewed as prime sources of information, due to their perceived competence and reliability of the information provided. Pharmacists, for example, were frequently consulted for their knowledge of what medications to take in case of general illness, and this translated into women's pursuit of information about their options for abortion care. For example, one woman who was seeking medication abortion shared, *"I didn't know where to start. I had gone to the pharmacy where they gave me a product . . .They told me that you drink one pill and the other you put [into your*

*vagina]. And, at night you will see the blood that will start to flow. There, you will now take anti-biotics to be taken afterwards."* (33-year-old woman, married, one child).

## Quality of interactions

The nature and quality of the information women received varied across sources and was as much about the decision to abort as it was about the methods and risks involved. Some exchanges were aimed at discouraging women from terminating the pregnancy, while other conversations focused on the methods and safety of the procedures, warning of the risks and perceived negative consequences of abortion, including inaccurate information.

## Attempts at deterrence

In seeking advice on where and how to obtain an abortion from those around them, many women were discouraged from terminating the pregnancy. Sometimes this took the form of a partner's refusal or a family member's fear of tarnishing the family's honor due to the stigma of abortion in the community.

As one adolescent shared, *"On the same day of the abortion at home, my mother was already prejudiced and indirectly forbade me to be able to have an abortion by saying, 'At home, here, none of my children will be able to have an abortion, and the one who will dare to do it will die.' Afterwards, my older brother, who lives in Gombe, called me and said, 'I just had a dream that you are having an abortion, while I never have an abortion and I don't want one of my sisters to have one.' I was so afraid."* (18-year-old woman, never married, no children).

Some women expressed how they were discouraged from terminating pregnancies by their partners. In some cases, women acted independently in seeking the abortion, despite this attempt at deterrence; as another adolescent described, *"My partner had refused to have an abortion, but I had done it unilaterally. I could see my partner's state of mind was not good . . .But I didn't tell my partner because if and only if my partner knew about it, he didn't want me to have an abortion."* (19-year-old woman, never married, no children).

Some providers only offered medical advice without facilitating the abortion for fear of legal reprisal. However, several women seemed to overcome providers' reluctance by involving their male partners or offering providers bribes for their services. For example, one 20-year-old woman shared her experience of negotiating with the provider alongside her male partner, saying, *"[My boyfriend and I] went to a pharmacy to buy a medicine, whose name I forget, but the pharmacist refused to serve us because it is only for married people and boys. I bribed the pharmacist. . .and he served me but telling me that if this product has side effects on you it will have nothing to do with me."* (20-year-old woman, never married, no children). Others were less successful in their efforts to persuade providers, as one young woman described, *"Like when my boyfriend and I went to see the first doctor, at first, he didn't agree. He advised against it because he felt that after an abortion, there are always negative repercussions that could even cause him to stop or even close his hospital. He really advised against it and said: 'It's better to have a miscarriage than an abortion'."* (18-year-old woman, never married, no children).

## Heterogeneous information on procedures

The informant network was a key determinant of the quality of the information that women obtained. While providers generally advised recommended methods, this was often not the case with social network or community sources, which relayed information on non-recommended methods, such as commonly used drugs or traditional or home remedies that frequently led to complications.

Media was rarely mentioned as a source of abortion information (and the internet or social media were never mentioned), providing only sporadic and incomplete exposure to abortion methods, as one participant shared, *"I had pretty poor knowledge about abortion, but on TV I had once heard about curettage and abortion pills."* (40-year-old woman, living with partner, three children).

Others who relied on their friends or community network similarly received incomplete and inaccurate information about abortion options, often resulting in use of non-recommended methods. One woman described recommendations she received to consume an herbal remedy, stating, *"Yes, I already knew that you can take the 'tangawisi', and it can remove the pregnancy. From what is said in the street by everyone, a pregnant woman cannot drink the tangawisi juice, 'esopaka zemi' (i.e., this juice has 'abortive virtues'). From such information, I had known that if I drink the tangawisi, my pregnancy will go away."* (29-year-old woman, never married, two children). Scenarios like these were common when women sought information from members of their community, avoiding care and clinical guidance from healthcare providers.

In contrast, conversations with healthcare providers generally focused on the abortion method deemed appropriate according to the gestational age of the pregnancy. These interactions also included information about the safety of the procedure by explaining possible complications. However, the complications were often overestimated or misrepresented compared to the risks of the WHO-recommended methods. The risks appeared to be amplified as a mechanism for dissuading women or reflected the reality of a context characterized by suboptimal quality of care.

As two respondents shared their positive care experience with a provider: as one woman shared, *"He [the provider] had explained everything to me, the pros and cons of abortion."* (27-year-old woman, never married, one child), and another, *"When he [the doctor] came to the house I asked him questions, I asked him if we should go through curettage or other methods, he told me that we would buy the medicines and that I would take them. He told me that I was going to have excruciating pain for 3 or 4 days because it was a forced thing. . .he explained everything to me. He told me about all the complications that could occur, such as damaging the woman's cervix, even getting to the point of cutting the tubes, not having any more children. He made me see all the possible consequences."* (21-year-old woman, never married, one child). Similarly, one participant described how her interaction with the health provider emphasized potential long-term consequences of an abortion for her fertility, which were not grounded in evidence, *"He [the nurse] told me that I can be at risk. It can have complications such as infertility."* (23-year-old woman, living with partner, one child).

Overall, women relied heavily on information circulating in their communities about their methods and sources for seeking an abortion. While women shared their experiences of successfully obtaining information from the formal health sector, guidance from providers was generally of low quality, as many evidence-based aspects of safe abortion care were not shared with women or were communicated inaccurately.

## Discussion

Findings from this qualitative study among women in the DRC illustrate the complex ways in which women pursue information to navigate their options for safe abortion care. In the context of DRC's recently liberalized abortion laws, this study highlighted 1) how the need to maintain social safety drives a selective and narrow information-seeking processes involving very few (if any), trusted individuals, and 2) how quality of interactions varied from overt

attempts at deterrence or receipt of inadequate information to more medically accurate information, depending on source.

Fear of social stigma and rejection shaped participants' pathways for information-gathering, echoing findings from other contexts [15–17]. These concerns reflect a reality for many women, who are often ostracized, isolated, and sometimes forced to leave their communities if they are found to have had an abortion [23–25]. While the present study is unable to evaluate how the change in the legal status of abortion in 2018 may have impacted information seeking, we anticipate the fear of social sanctions remains a major concern, especially given 19% of women in Kinshasa were unaware of the legal conditions for abortion in 2021, three years after the legal change [11]. To avoid these social sanctions, women were highly selective about to whom they disclosed their pregnancies, often describing the source of their trust and confidence in their selected informant(s). While several studies link fear of stigma to limited pregnancy disclosure [19, 26, 27], women's decision-making in these scenarios prioritized privacy even in the absence of complete or accurate information about abortion options. These results suggest the need for public awareness campaigns and community-level outreach to inform people about the risks of unsafe abortion and the availability of legal reproductive health services to reduce these risks, which affect many families in the community. Beyond awareness campaigns, efforts should address the stigmatization of sexual and reproductive health services, including abortion care, at the community level to improve women's access to safe abortion care. One of the ways women seek information while avoiding public exposure is to involve their partners who do not face the same risk of social shame. As suggested in other studies conducted in legally restricted abortion settings [12, 28], we found male partners take on much responsibility in women's abortion trajectories, often connecting their pregnant partners directly to healthcare providers. However, in doing so, our results indicate some partners infringed on women's reproductive autonomy, making decisions without sharing information.

Altogether, few women in our study knew about safe options to terminate their pregnancy, which corroborates the quantitative results showing only 23% of women were aware of a recommended method for abortion [11]. Even in case of interactions with healthcare providers, our study revealed low-quality advice and information from healthcare providers. This may be explained by the lack of comprehensive, woman-centred abortion care and inadequate training in recommended abortion methods [4, 29–31].

Other research has shown low levels of medication abortion knowledge and incomplete information provided to women among pharmacy providers or drug sellers, which is consistent with our findings [29, 32, 33]. These results highlight the need to monitor the implementation of legal extensions to safe abortion care in the DRC, given concerns about low quality of care received. This is consistent with recent findings suggesting inadequate safe abortion care service availability and readiness to provide quality care in 2021 in Kinshasa [31]. In addition, the provision of legal abortion care should consider task shifting opportunities, including the training of pharmacist who are among the most popular contact points for women seeking information on abortion care. The training of mid-level providers and pharmacists is in line with current WHO guidelines on safe abortion care and has the potential to significantly increase the quality of safe abortion care, thereby reducing abortion-related complications.

This study is not without limitations. First, our qualitative interview guides were developed to capture a range of narratives and experiences related to abortion care-seeking, and, thus, did not focus exclusively on the information-gathering process that this study explores. While multiple questions in our interview guide asked women who they engaged with to seek information about their options for abortion, additional information about the sequence, timing, and consequences of information-seeking experiences may have been garnered through dedicated

questions on this topic. Second, interviewers were trained to build rapport with and support women in sharing their experiences, yet the stigmatizing nature of abortion in DRC may have inhibited some women from divulging certain details about who they engaged and why.

Despite these limitations, our study has many strengths. This includes the large number of qualitative interviews conducted with a diverse population of women who had an abortion in the last decade. We leveraged an existing population-based survey measuring abortion incidence and safety to identify eligible women for our qualitative study, reducing potential bias that could be introduced through other recruitment methods (e.g., facility-based) and increasing the representation of the diverse population of women who have abortions in the DRC. Our multi-country research term, including researchers from Kinshasa, the DRC, and the US, collaborated throughout the research process to ensure interviews, coding, and analysis were conducted consistently across team members and results were interpreted with nuanced understanding of the study context.

Women in the DRC are highly selective in how they seek information about their options for abortion care, favoring their close social network rather than professional support for fear of social sanctions. As a result, women typically rely on insufficient and often inaccurate information to make abortion care decisions, resulting in an unnecessarily high risk of complications from unsafe abortion methods. While legal actions to improve access to safe abortions are essential, the public health gains will not materialize unless legal action is enhanced by community-level actions to raise awareness about the legality and availability of safe abortions services, as well as medications available for safe self-managed abortion.

## Author Contributions

**Conceptualization:** Denise Ngondo, Celia Karp, Dynah Kayembe, Caroline Moreau, Pierre Akilimali, Suzanne O. Bell.

**Data curation:** Denise Ngondo, Celia Karp, Dynah Kayembe, Kisulu Samyonga Basile, Pierre Akilimali, Suzanne O. Bell.

**Formal analysis:** Denise Ngondo, Celia Karp, Dynah Kayembe, Pierre Akilimali.

**Funding acquisition:** Pierre Akilimali, Suzanne O. Bell.

**Investigation:** Denise Ngondo, Celia Karp, Dynah Kayembe, Kisulu Samyonga Basile, Caroline Moreau, Pierre Akilimali, Suzanne O. Bell.

**Methodology:** Denise Ngondo, Celia Karp, Dynah Kayembe, Kisulu Samyonga Basile, Caroline Moreau, Pierre Akilimali, Suzanne O. Bell.

**Project administration:** Denise Ngondo, Dynah Kayembe, Kisulu Samyonga Basile, Pierre Akilimali.

**Resources:** Suzanne O. Bell.

**Supervision:** Pierre Akilimali.

**Validation:** Denise Ngondo.

**Writing – original draft:** Denise Ngondo, Celia Karp, Caroline Moreau, Suzanne O. Bell.

**Writing – review & editing:** Denise Ngondo, Celia Karp, Dynah Kayembe, Kisulu Samyonga Basile, Caroline Moreau, Pierre Akilimali, Suzanne O. Bell.

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
