## [Decision Letter · Decision Letter 0]

26 Jun 2023

PGPH-D-23-00900

Abortion information-seeking experiences among women who obtained abortions in Kinshasa, DRC: results from a qualitative study

Dear Dr. Karp,

Thank you for submitting your manuscript to PLOS Global Public Health. After careful consideration, we feel that it has merit but does not fully meet PLOS Global Public Health’s publication criteria as it currently stands. Therefore, we invite you to submit a revised version of the manuscript that addresses the points raised during the review process.

We look forward to receiving your revised manuscript.

Kind regards,

Olivia Miu-yung Ngan

Academic Editor

Journal Requirements:

2. We have noticed that you have uploaded Supporting Information files, but you have not included a list of legends. Please add a full list of legends for your Supporting Information files after the references list. 

Additional Editor Comments (if provided):

1. In the abstract, spell out DRC before the use of the abbreviation

2. In the introduction paragraphs

a. Any time restriction on abortion?

b. What is the primary access to abortion care in Kinshasa?

c. How much does it cost? Is it affordable to most women, or is it privileged access?

d. The authors mentioned the stigmatisation around abortion. Are any cultural elements specific to the DRC population?

e. Does the decriminalisation of abortion reduce the number of unsafe abortions? What is the number of unsafe abortion care since the decriminalisation?

f. What prompts people to use unsafe options before and after decriminalisation? What are the barriers?

g. What indicators does the legalisation of abortion bring public health good?

h. What is the source of information before and after the decriminalisation?

3. Methods:

a. Remove the sentence, “I thought it was mentioned. It not, this is from…..”.

b. Have the survey results been published? Citation is missing.

c. How did the research team select to contact 65 women from the 93 women who reported having had an abortion between 2015 and 2021? What were the inclusion and exclusion criteria?

d. Line 131- 132. The authors mention the four-day training about research ethics, informed consent etc. Of the nine interviewers, what are their prior backgrounds? Any research exposure or community background?

e. Interviews were conducted in what language? French?

f. Ethical Concern – some women mentioned their use of abortion, even if it is not legalised, during the interview. How are women be protected from this research?

4. Results and Discussion:

a. Have any interviewees obtained more than one abortion in their lifetime? It is not mentioned in the table.

b. Among interviewees who have obtained abortion before and after the legalisation, have they had different lived experiences regarding obtaining information or care?

c. What are the typical family dynamics in obtaining abortion care? Open disclosure? Hiding?

d. A rather large percentage of interviewees are never married and obtain an abortion. Any cultural or religious factors to this vulnerable population?

e. The authors provided many findings covering multiple themes, social networks and capitals to obtain information, stigma, and heterogeneous procedure information.

f. Discussion, however, is somewhat thin without an in-depth extrapolation. Authors are suggested to expand the elaboration by 1-2 paragraphs.

Reviewers' comments:

Reviewer's Responses to Questions

**Comments to the Author**

1. Does this manuscript meet PLOS Global Public Health’s publication criteria? Is the manuscript technically sound, and do the data support the conclusions? The manuscript must describe methodologically and ethically rigorous research with conclusions that are appropriately drawn based on the data presented.

Reviewer #1: Yes

Reviewer #2: Yes

2. Has the statistical analysis been performed appropriately and rigorously?

Reviewer #1: N/A

Reviewer #2: N/A

3. Have the authors made all data underlying the findings in their manuscript fully available (please refer to the Data Availability Statement at the start of the manuscript PDF file)?

Reviewer #1: No

Reviewer #2: Yes

4. Is the manuscript presented in an intelligible fashion and written in standard English?

Reviewer #1: Yes

Reviewer #2: Yes

5. Review Comments to the Author

Reviewer #1: I would like to thank the editor and PLOS Global Public Health for the opportunity to review this manuscript. The authors examined how women obtained information about abortion options in Kinshasa, DRC, by analyzing qualitative data collected from interviews. Key themes they found were 1) selective and narrow information search from the restricted range of their social network; and 2) heterogeneous information and stigmatizing interaction during the information search process. This study holds great significance as it contributes to generating high-quality evidence on the barriers and facilitators of accessing safe abortion, which is particularly relevant in light of recent policy changes and public debates taking place worldwide. The manuscript demonstrates strength in its methods and is effectively written. I have included specific questions and suggestions for the authors to consider.

Major comments

1. Methods, page 6, lines 119-120: The authors need to provide references and details to support their definition of recent abortion as reported abortion within the seven years prior to the survey. Please clarify the evidence or rationale behind this definition.

2. While the liberalization of abortion laws in 2018 is a significant event in the study setting, the study results lack adequate contextualization. As the recent abortion was defined as abortion that occurred between 2015 and 2021 (recent seven years), the authors may leverage the pre-post setting. It would be interesting to see the differences in information-seeking practices before and after the abortion liberalization in 2018. Media exposure surrounding the legal changes may have influenced women's awareness of abortion options and sources of information. This context needs to be manifested in their analysis to identify how women who seek information for abortion in DRC understand and interpret the context.

3. The authors identify two themes and then several subthemes under each, but they could have provided a more in-depth understanding of the relationships between the sub-themes. Under “Role of women in one’s community” (pages 14-16), the final paragraph (lines 330-337) does not seem to be about the role of other women in their social network. In addition, too many points are under this single subtheme (Role of women in one’s community), which could have been separated out and organized. I would suggest separating items under “Role of women in one’s community” and reorganizing them. For example, how the women utilize trusted people in their community or informal network (e.g., sisters, friends, and some other mature people who they could rely on) may be one subtheme, and then another subtheme may be the challenges and adverse consequences of the information seeking from the social network.

In addition, seeking information from healthcare providers may be a different theme. It is under the “Activation of the social network,” but it does not seem to be a social network strategy. Instead, it could be seeking information from formal resources, which may or may not be helpful. The challenges and barriers these women faced seem to be another important layer that should be noted because even women who seek complete and accurate information from healthcare providers might not obtain it due to more structural limitations.

4. Although low-quality guidance and information from providers emerged from their analyses, the Discussion section lacks substantial discussion on this matter. It would be helpful to discuss the quality of care, abortion providers’ behaviors as a trusted source of information and guidance of safe abortion care, and the practical implications arising from these findings.

5. The authors briefly mentioned media on page 19, lines 397-401, but did not elaborate further. Even in the first theme, there was no discussion on media. Was media (traditional form or social media) not an important source of information in this study setting? It would be helpful for readers to add some details about the frequencies of different information sources in the participant characteristics section.

Minor comments

1. Page 5, lines 108-109: There are unnecessary sentences. Please proofread the manuscript before submission.

2. Page 8. Line 168: It appears that Appendix Table 1 is not on the supplement materials list. Please provide the necessary information.

3. Page 9, Table 1: Education level would help readers better understand the study sample. In addition, I suggest classifying the abortion year based on the year of abortion liberalization in DRC.

Reviewer #2: Review comments

Thank you for the opportunity to review this manuscript. This is a timely paper which has the potential to shape policy on improving access to accurate abortion information and services in DRC.

Abstract

Line 28 _Authors may need to write out DRC in full before using abbreviation.

Introduction

Lines 59-60 _Is this referring to unsafe abortion practices? If yes, I would recommend authors use unsafe abortion practices for the sake of consistency. The author should kindly define what the recommended methods are, so that readers can appreciate the non-recommended methods.

Is there data on the prevalence of knowledge of the recent liberalized abortion law among women in DRC? The background might benefit from this data, and how that may be impacting on abortion seeking information.

Methods

The authors suggest that the study population was heterogeneous, but was saturation assessed in this study? Could the authors add a sentence or two on how saturation was tested and what was found?

Results

Lines 420-432_ Could these also be termed as attempted deterrent from health providers?

Discussion

The role of social stigma on participants’ pathways for information-gathering is critical as highlighted by the authors in lines 449-450. However, the authors concluding remarks do not contain any recommendations on how social stigma could be addressed even if there was high awareness about abortion legality in DRC and availability of safe abortions services at the community level.

6. PLOS authors have the option to publish the peer review history of their article (what does this mean?). If published, this will include your full peer review and any attached files.

**Do you want your identity to be public for this peer review?** For information about this choice, including consent withdrawal, please see our Privacy Policy.

Reviewer #1: No

Reviewer #2: **Yes: **Maxwell Tii Kumbeni

---

## [Decision Letter · Decision Letter 1]

27 Dec 2023

Abortion information-seeking experiences among women who obtained abortions in Kinshasa, DRC: results from a qualitative study

PGPH-D-23-00900R1

Dear Dr. Karp,

We are pleased to inform you that your manuscript 'Abortion information-seeking experiences among women who obtained abortions in Kinshasa, DRC: results from a qualitative study' has been provisionally accepted for publication in PLOS Global Public Health.

Best regards,

Olivia Miu-yung Ngan

Academic Editor

Reviewer Comments (if any, and for reference):

Reviewer's Responses to Questions

**Comments to the Author**

1. If the authors have adequately addressed your comments raised in a previous round of review and you feel that this manuscript is now acceptable for publication, you may indicate that here to bypass the “Comments to the Author” section, enter your conflict of interest statement in the “Confidential to Editor” section, and submit your "Accept" recommendation.

Reviewer #3: (No Response)

Reviewer #4: All comments have been addressed

2. Does this manuscript meet PLOS Global Public Health’s publication criteria? Is the manuscript technically sound, and do the data support the conclusions? The manuscript must describe methodologically and ethically rigorous research with conclusions that are appropriately drawn based on the data presented.

Reviewer #3: Yes

Reviewer #4: Yes

3. Has the statistical analysis been performed appropriately and rigorously?

Reviewer #3: N/A

Reviewer #4: N/A

4. Have the authors made all data underlying the findings in their manuscript fully available (please refer to the Data Availability Statement at the start of the manuscript PDF file)?

Reviewer #3: No

Reviewer #4: Yes

5. Is the manuscript presented in an intelligible fashion and written in standard English?

Reviewer #3: Yes

Reviewer #4: Yes

6. Review Comments to the Author

Reviewer #3: Data availability:

According to the PLOS Data Policy “When specific legal or ethical restrictions prohibit public sharing of a data set, authors must indicate how others may obtain access to the data.”. This is missing, please indicate how others may obtain access to the data used in this study.

Methods:

Defined initially as in-depth interviews (line 130) and later on as structured interviews (line 138). In-depth interviews is a method characterized differently from structured interviews. Instead of talking about ‘in-depth interviews’ it would more clear to the reader to read ‘interviews conducted in depth’ (if this is what the authors mean). Or are these interviews more of a semi-structured interviews?

Line 139-140: “The guide was pre-tested during the interviewer training”. Please specify with whom (interviewee) it was pre-tested? With women from the sample? Comparable women? Among the interviewers?

Results:

Please amend typos in Table 1 in Results:

- ‘53.,9’ instead of ‘53.9’.

- ‘*: One woman’ instead of ‘*One woman’

In the sub-section Activation of the social network, no quote from interviewees’ accounts has been quoted. Quotes are desirable to justify your conclusions in the analysis.

Potential inconsistency or translation mistake: line 278-279 quotes the interviewee talking about her spouse but in line 281 this interviewee is defined as ‘never married’.

Please make sure all interviewees are described in the same manner when possible. F.i. line 335 ‘(Never married woman, no children)’ is missing the age information.

In line 512-513 authors acknowledge the effect of stigmatization in DRC in inhibiting some women from given details. A potential major limitation is the sex of the interviewers. This can be acknowledge by firstly, providing in the methods the number of male and female interviewers ,and secondly, reflecting in this paragraph on the potential effect that this male-female distribution can have.

Please amend typo in line 521 ‘research term, including researchers from' instead of ‘research team’

Reviewer #4: Thank you for your insightful and well researched manuscript. I was particularly impressed by the thoroughness with which you addressed the reviewer feedback, and I believe that your work makes a significant contribution to our understanding of women's abortion information seeking experiences in the Democratic Republic of the Congo (DRC).

7. PLOS authors have the option to publish the peer review history of their article (what does this mean?). If published, this will include your full peer review and any attached files.

**Do you want your identity to be public for this peer review?** For information about this choice, including consent withdrawal, please see our Privacy Policy.

Reviewer #3: No

Reviewer #4: No
